# Mapping current and future flood exposure using a 5-metre flood model and climate change projections

Connor Darlington[1], Jonathan Raikes[1,2], Daniel Henstra[3], Jason Thistlethwaite[1], Emma K. Raven[4]

[1]School of Environment, Enterprise and Development, University of Waterloo, Waterloo, Ontario, N2L 3G1, Canada
[2]Sustainability Research Centre, University of the Sunshine Coast, Sippy Downs, Queensland, 4556, Australia
[3]Department of Political Science, University of Waterloo, Waterloo, Ontario, N2L 3G1, Canada
[4]JBA Risk Management, Broughton, Skipton, BD23 3FD, United Kingdom

*Correspondence to*: Jonathan Raikes (jraikes@uwaterloo.ca)

**Abstract.** Local stakeholders need information about areas exposed to potential flooding to manage increasing disaster risk. Moderate and large-scale flood hazard mapping is often produced at a low spatial resolution, typically using only one source of flooding (e.g., riverine), and it often fails to include climate change. This article assesses flood hazard exposure in the City of Vancouver, Canada, using flood mapping produced by flood risk science experts JBA Risk Management, which represented baseline exposure at 5-metre spatial resolution and incorporated climate change-adjusted values based on different greenhouse gas emission scenarios. The article identifies areas of both current and future flood exposure in the built environment, differentiating between sources of flooding (fluvial, pluvial, storm surge), and climate change scenarios. The case study demonstrates the utility of a flood model with a moderate resolution for informing planning, policy development, and public education. Without recent engineered or regulatory mapping available in all areas across Canada, this model provides a mechanism for identifying possible present and future flood risk at a higher resolution than is available at Canada-wide coverage.

**Keywords.** Flood hazard mapping; flood risk management; exposure; climate change; flood modelling

**Plain Text Summary.** The impacts of climate change on local floods require precise maps that clearly demarcate changes to flood exposure; however, most maps lack important considerations that make their utility in policy and decision-making difficult. This article presents a new approach to identifying current and projected flood exposure using a 5-metre model. The results highlight advancements in the mapping of flood exposure with implications for flood risk management.

## 1.0 Introduction

The exposure of people and infrastructure to flood hazards is increasing globally, due to factors such as population growth, development in flood-prone areas, and more frequent and intense extreme weather caused by climate change (Field et al., 2012; UNDRR, 2022). Moreover, it is expected that all major types of flooding, including fluvial

(riverine), pluvial (rainfall) and storm surge (coastal) will intensify as the climate changes (Alfieri et al., 2016; Arnell

& Gosling, 2016; Hirabayashi et al., 2021; IPCC, 2019; Muis et al., 2016; Winsemius et al., 2016).

Coastal cities are especially susceptible to flooding, due to their dense populations, socio-economic development, impervious surfaces, and proximity to major hydrological features such as lakes and oceans (Hallegatte et al., 2013; Lincke et al., 2022; McDermott, 2022; Neumann et al., 2015). Managing flood risk in coastal cities requires adopting

actions that reduce the vulnerability of people and property to current flood hazards, but also to anticipate the likely scope and extent of future flooding. Flood hazard modeling and mapping that uses climate scenarios to estimate future flood exposure enables coastal cities to better support flood risk management, inform land use planning, organize emergency management, and increase public awareness (Dransch et al., 2010; Handmer, 2013; Porter & Demeritt, 2012).


Despite the importance of flood hazard mapping for flood risk management, few studies have mapped community exposure to multiple flood types and used future climate scenarios to assess changes to exposure (Cea & Costabile, 2022). Modeling techniques to estimate flooding under different climate scenarios vary considerably in existing scholarship, and the quality and granularity of local and regional flood maps are also highly variable (Cea & Costabile,

2022; Costabile et al., 2015; de Moel et al., 2009; Henstra et al., 2019; Mudashiru et al., 2021). These limitations underscore the need to develop flood hazard models and maps that capture flood exposure accurately and at a resolution that is useful for planning and decision-making.

The purpose of this paper is to expand on traditional physically-based flood exposure modeling and mapping

methodologies that often lack consideration of multiple flood mechanisms and climate change at a high resolution. This paper presents the results of a flood model that was used to produce flood hazard maps under various climate change scenarios for the City of Vancouver, Canada. Using a 5-metre resolution baseline and climate change-adjusted flood data produced by flood risk science experts at JBA Risk Management (JBA), we determined areas of existing building exposure to multiple flood types, as well as new exposure based on climate change scenarios for 2050 and

2080. The findings demonstrate the utility of local and regional flood exposure analysis using different climate change scenarios, which offers guidance for local planners, policy- and decision-makers, and other stakeholders to recognize areas of current and future flood risk and enact measures to manage this risk.

The paper begins by reviewing current scholarship on flood hazard mapping to distinguish different methodologies,

assess their applicability in Canada and beyond, and identify knowledge gaps. It then describes the flood hazard mapping approach used in this study and its application in Vancouver. The third section reports the study's main findings. The paper concludes with a broader discussion on the strengths and limitations of the method, directions for its use in local and regional planning, and areas for future research.

**2.0 Literature Review: Flood Hazard Mapping Methodologies**

Scholarship on flood hazard mapping has been increasing for decades. Early approaches to flood hazard modeling were incapable of incorporating long-term climate projections and variations to hydrological processes (Batista, 2018). More contemporary approaches rely on computer modeling and mapping that can apply scenario-based projections of climate change and precipitation (Mudashiru et al., 2021; Teng et al., 2017). This section reviews current scholarship on methodologies for modeling flood exposure and its application in Canada with climate change.

**2.1 Modeling Flood Exposure**

There are three main methodologies for producing flood hazard maps, which include physical modeling, physically-based modeling, and empirical modeling (Mudashiru et al., 2021; Teng et al., 2017). Physical models map flood hazards using field measurements and observations of hydrological features, such as the velocity and flow of a meandering river (Mubialiwo et al., 2022; Paquier et al., 2017). Physical models produce the most accurate and highest

resolution picture of flood hazards (e.g., 1-metre), but the on-site measurement and testing requirements are onerous, time-consuming, and costly, such that these models are typically limited to a small spatial coverage (Bellos, 2012).

Physically-based models simulate real-world hydrological processes to identify areas that could be inundated under various conditions (e.g., extreme weather, riverine flow patterns) (Mudashiru et al., 2021). These models are

increasingly relied on by flood risk management practitioners because of their capacity to integrate 1-, 2-, and 3-dimensional hydrodynamic models (Anees et al., 2016). 1D models are mostly adopted for river studies when the computational requirements required to estimate flood exposure are more limited (Horritt & Bates, 2002; Dazzi et al., 2021). Meanwhile, 2D approaches are becoming more common in the field of flood hazard modelling and mapping due to increased access to Digital Terrain Models for geographically large areas (Dazzi et al., 2021). However the

computational requirements to run sophisticated flood models using 2D hydrodynamic models over a large area are significant. In data-sparse regions and areas where data acquisition and procurement are more limited due to administrative constraints, access to high-quality 2D approaches may be hindered. Similarly, 3D models provide a more nuanced examination of flood hazard exposure, but the computational requirements to run such a model are prohibitive map the results requires specialized technical expertise. Overall, physically-based models reduce the need

for field observations, which are instead simulated in a lab, enabling researchers to extrapolate field observations to cover a larger area in less time. However, the accuracy of these maps is sometimes challenged by critics who question assumptions about the hydrological processes that have not been fully tested in the field (Costabile et al., 2015; Mark et al., 2004).

Empirical modeling is a more recent development in flood hazard assessment that typically combines satellite imagery, remote sensing, machine learning, artificial intelligence, and geographic information systems to predict areas exposed to flood inundation (Devia et al., 2015). This approach has become more common in conventional flood hazard mapping because it can produce maps with large spatial coverage, it is less onerous and more efficient than physical and physically-based models from a cost-benefit perspective, and it is capable of incorporating environmental changes

such as those associated with climate change (Jehanzaib et al., 2022; Mosavi et al., 2018; Mudashiru et al., 2021). However, empirical models typically produce lower-resolution maps (e.g., 30-metre or lower) and make broader assumptions about physical conditions than physical and physically-based models (Avand et al., 2022; Li et al., 2013; Woznicki et al., 2019).

Physically-based maps producing high levels of accuracy tend to be costly and require significant resources and time, whereas empirical models are less accurate but require less resources and can be deployed at a broader scale. Policymakers must assess these trade-offs when determining which maps should be generated for specific locations and audiences. For example, small and remote communities might lack the financial capacity to conduct physical modeling, so a cost-efficient, multiple return period model might be desirable, as it can map hazard exposure and

incorporate climate change projections at a moderate resolution that is sufficient for planning and decision-making. For this reason, this study used a physically-based modeling approach as a sensible middle ground and starting point. The next section describes the evolution of flood mapping in Canada and how maps are used in flood risk management.

**2.2 Modeling Canada's Flood Exposure Under Climate Change Scenarios**

Despite the value of flood hazard maps for land use planners, emergency managers, and other stakeholders, several

factors limit their utility in practice. In particular, many existing flood hazard maps lack high-resolution data, fail to represent multiple sources of flooding (e.g., fluvial, pluvial, and storm surges), and neglect to incorporate the influence of climate change on flood exposure (Cea & Costabile, 2022; de Moel et al., 2009; Teng et al., 2017). Moreover, the variable accuracy and reliability of flood hazard maps produced through different modeling approaches, often with different assumptions and using coarse resolution data, as well as the technical and financial requirements to produce

higher-quality maps, often hinders their availability and effective use by non-expert stakeholders, such as planners and policymakers (Dransch et al., 2010; Hagemeier-Klose & Wagner, 2009; Pralle, 2019; Wing et al., 2018).

Flood hazard mapping in Canada is highly variable, due in part to the country's large geographic area and diverse topography (Elshorbagy et al., 2018). Flood mapping is a provincial and territorial responsibility, with some provinces

and territories performing mapping in-house, while others contract flood mapping to private industry (Natural Resources Canada, 2022a). Because provincial governments have primary responsibility for flood hazard mapping, there is a patchwork of coverage and map availability across Canada. Further, despite recent data initiatives to compile flood data (Natural Resources Canada, 2023), there is no national, high-resolution physical modeling for all of Canada.

There are significant gaps in flood mapping coverage across Canada, and the dominant focus of nearly all regulatory flood mapping is fluvial (riverine) flooding. To date, physically-based flood hazard modeling has been relatively unavailable in Canada, particularly modeling that includes widespread coverage, captures multiple sources of flooding, and accounts for climate change (MMM Group Limited, 2014). However, few commercial risk modeling companies offer solutions with national or near-national coverage that include areas not otherwise mapped in Canada.

Whereas organizations such as the First Street Foundation and Fathom offer comprehensive coverage of the United

States and elsewhere, no such modelling in Canada has been completed. To accomplish such large-scale modeling, considerable climate modeling, hydraulic modeling, data collection, and computational resources are required. Such models can be a useful source of flood intelligence as computational and physically-based modeling improve in accuracy with advancements in data availability and computational resources.


Against this backdrop, we present here an assessment of flood exposure using JBA's physically-based flood hazard model at a 5-metre resolution that includes multiple flood types—fluvial, pluvial and storm surge—and estimates future changes due to climate change based on the Representative Concentration Pathways (RCP) 4.5 and 8.5 climate scenarios. We apply this model using a case study of the City of Vancouver, British Columbia, which faces risks from

all three flood types. This study illustrates a practical application of physically-based modeling for the purposes of generic flood exposure assessment and flood risk planning.

**3.0 Methods**

This section describes the methods used to harness the physically-based flood hazard model and its application to the City of Vancouver. It includes an overview of the study area and research methods that were used to assess the current

and projected changes to local flood exposure based on climate change scenarios, multiple sources of flooding, and time horizons.

**3.1 Study Area**

The scope of this assessment was limited to the JBA Canada 5-metre Baseline and Climate Change Flood Data study site for the Vancouver area (JBA Risk Management Limited, 2022). This dataset contained fluvial, pluvial, and storm

surge flood hazard data under non-climate change (NCC) and climate change states, specifically RCP 4.5 and RCP 8.5 climate scenarios for 2050 and 2080. The flood hazard data was produced by JBA and shared with the University of Waterloo research team for the Vancouver metropolitan area. This area delineates 117 contiguous census tracts in the city from the 2016 open census tract boundary file (Statistics Canada, 2019). A breakdown of the fluvial, pluvial, and storm surge flood mechanisms in Vancouver is provided in Figure 1 at the non-climate change (NCC) state and

at the 100-year return period, utilized for illustrative purposes.

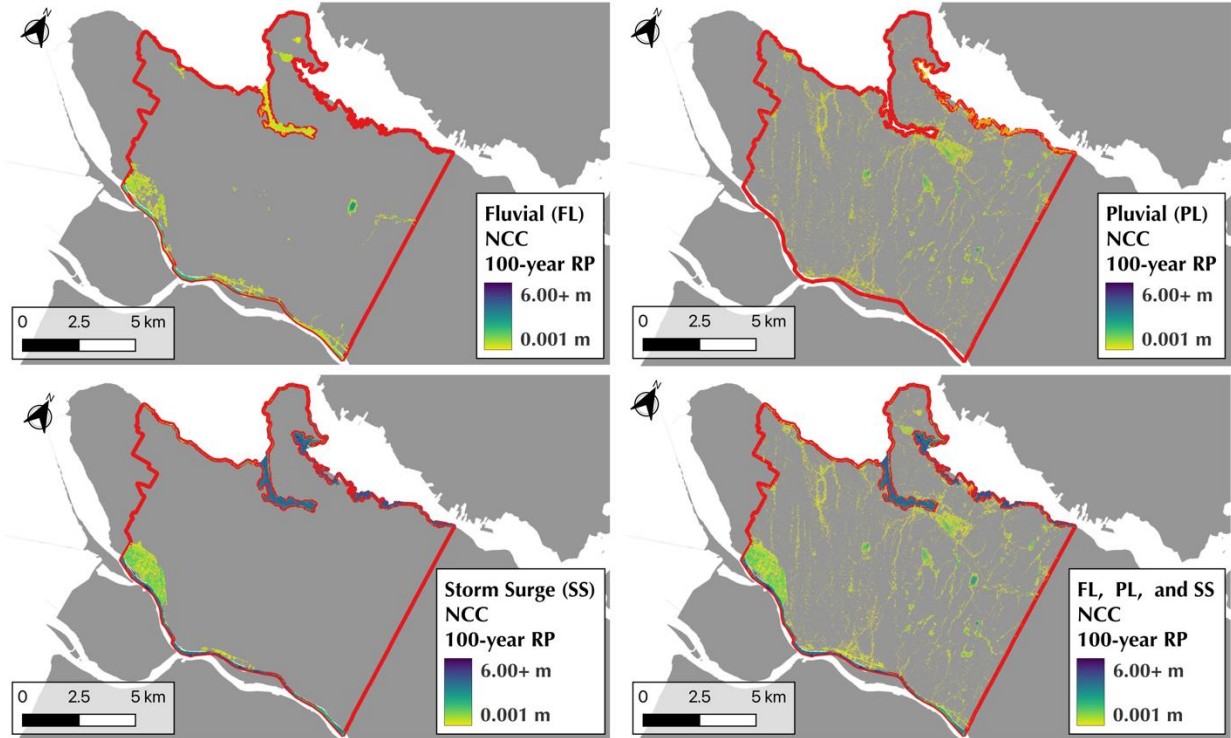

**Figure 1: Study boundary of the JBA (2022) flood hazard data for Vancouver, British Columbia for the fluvial, pluvial, and storm surge flood hazard modeling at the 100-year return period non-climate change state. All three flood mechanisms are provided in the same map to illustrate the overall flood exposure independent of flood mechanism. Other map data includes the Statistics Canada (2022) provinces and territories boundary file of Canada.**

The study area illustrated in Figure 1 is used for the remainder of the assessment. An example exposure data set provided by Microsoft (2019) was incorporated to indicate the types of analysis that could be conducted using the JBA data.

### 3.2 Research Methods

This study is based on JBA's Canada 5-metre Flood Data, baseline and future (and Canada-wide 30m flood hazard data), obtained through a data sharing agreement with the University of Waterloo. JBA's in-house two-dimensional hydrodynamic flood model, JFlow (see Lamb et al., 2009 for more information) was used to map fluvial and pluvial flood extents and depths. JFlow solves the shallow water equations to simulate flooding and is configured differently to generate the pluvial and fluvial flood maps (e.g. modeling along a river network, or across pluvial rainfall catchments). The modeling was performed on the best available terrain data which included 1m LiDAR in the Vancouver urban area. The terrain data were used to derive river locations and catchment boundaries and were processed and edited to improve quality (e.g. by removing structures such as bridges which block the natural flow of water). The hydrological inputs required by the hydraulic model vary for different configurations. Depending on

catchment size, separate model set ups were used to represent the different way small and large rivers respond to storm events.


Rivers draining areas greater than 400km$^2$ were classed as 'large rivers' and were modeled to create the fluvial flood maps. For these rivers, hydrographs for each return period being modeled were derived from a statistical analysis of flood peak gauge data extrapolated from the Water Survey of Canada's HYDAT database (Environment and Climate Change Canada, 2018). To perform this analysis, the median annual maximum flood (QMED) was calculated,

representing the 2-year river flow return period. QMED was directly identified for gauged stations with a minimum data record of 10 years. At ungauged locations, QMED was statistically derived, accounting for regional and local climatic factors. After QMED had been calculated for all locations, this was scaled to generate peak flow data for the return periods required using flood growth curves. A flood growth curve describes the ratio between the QMED flow and those at other return periods. The impact of snowmelt was implicitly accounted for in the peak flow data analysis.

The design flood depths were turned into hydrographs that represent the volume of water through time and routed through JFlow.

Rivers draining less than 400km$^2$ were classed as 'small rivers and pluvial maps'. Small river and pluvial catchments are more responsive to highly localised, intense rainfall than large rivers; therefore, a different approach was applied

to capture the maximum likely flood hazard in these smaller catchments. The approach used, referred to as direct-rainfall modeling, estimates design rainfall hyetographs (the distribution of rainfall intensity over time). Rainfall intensity-duration-frequency statistics available from Environment and Climate Change Canada were used to interpolate rainfall estimate data for all catchments. A range of design storm durations were modeled to identify the critical rainfall duration, or maximum likely flood hazard, in each catchment. Short, intense rainfall events tend to

generate more flooding in steep-sided valleys, whereas flatter regions are often adversely affected by slower moving storms with longer storm durations. To account for this, rainfall totals for 1-hour, 3-hour and 24-hour storm durations were used to create multiple modeled flood extents per return period. The flood extent with the greatest water depths was then used for each modeled return period. These estimated rainfall data were used to generate storm hyetographs and an infiltration coefficient was applied to remove the proportion of rainfall that would infiltrate the ground due to

urban drainage, infiltration and interception. This varies across different land surfaces and climate types: defined using the eco-geographical divisions, called ecozones, from the National Ecological Framework of Canada (Agriculture and Agri-food Canada, 2023). For urban areas, a runoff value of 85% was applied (i.e. 15% loss). This value was selected based on the well-established US SCS curve number method (USDA, 1986b) where high numbers represent higher runoff (e.g. 98 for completely paved and 46 for low-density residential with permeable soil). After sensitivity testing

of curve numbers, an infiltration coefficient of 85 was selected as a single best value to represent Canadian urban areas. The impacts of seasonality, including the role of frozen ground and seasonal snowmelt were also accounted for in the small river and pluvial hydrology. A post-processing step was used to extract all the small rivers from the direct rainfall modeling and add them to the fluvial flood maps using the National Hydro Network.

For JBA's coastal flood maps, extreme sea-levels for a range of return periods were estimated using permanent tide gauge data from Fisheries and Oceans Canada and the U.S. National Oceanographic and Atmospheric Administration. In addition, JBA generated hindcast modeled water levels (using ADCIRC and TELEMAC-2D hindcast models) at locations between gauge sites to derive a complete set of extreme sea levels around the coastline. GIS horizontal projection modeling was used to determine the extent and depth of coastal flooding from these sea-level extremes
across the inland terrain data.

To produce climate change-adjusted flood mapping, JBA adjusted the input hydrology to reflect anticipated changes. By comparing the statistical differences between the baseline and future climate change scenarios, 'change factors' were calculated to quantify the measure of change. These change factors were then applied to the baseline hydrology
to create a new set of future inputs. The new inputs were run in JFlow to map future flood extents and depths. For pluvial flooding, precipitation intensity-duration-frequency (IDF) curves for present day and potential future scenarios at gauged and ungauged locations were obtained from Western University (Simonovic et al., 2023). Future precipitation depths and associated durations were divided by present day volumes to obtain the change factor, which was multiplied by the timesteps in the hyetograph to provide the future projected design hyetograph.

To calculate change factors for fluvial flooding, future projections of precipitation and temperature were obtained from the Climate Atlas (Prairie Climate Centre, 2022). Monthly adjustments were extrapolated to the daily scale and used to adjust rainfall-runoff data in JBA's baseline fluvial models to derive future estimates of river flow extremes. These were also modelled in JFlow to map new fluvial flood extents and depths under climate change. For the coastal
climate change estimates, sea-level rise information was obtained from the Canadian Extreme Water Level Adaptation Tool (Zhai et al., 2023) and future sea-level extremes were used to adjust coastal boundary conditions to map future coastal flooding.

Access to the 5-metre baseline and climate change flood map data was intended to pilot and explore the benefits of
higher resolution local flood hazard maps compared to the Canada-wide resolution which is traditionally 30-metre resolution. The data were provided as a series of raster files, with each file reflecting a return period, flood mechanism, and climate state. For example, one raster file consisted of the 100-year return period, fluvial-sourced, NCC flood hazard estimation. JBA data include three main sources of flooding: (1) fluvial, (2) pluvial, and (3) storm surge flooding, each at seven different return periods (Table 1).

**Table 1: Flood Hazard Data Provided by JBA**

| Return Period | Annual Exceedance Probability (AEP) |
|---|---|
| 20 | 0.05000 |
| 50 | 0.02000 |
| 75 | 0.01333 |
| 100 | 0.01000 |
| 200 | 0.00500 |

| | |
|---|---|
| 500 | 0.00200 |
| 1500 | 0.00067 |

Additionally, JBA developed climate change flood hazard estimations at the RCP 4.5 and RCP 8.5 scenarios. Two separate time periods of assessment were used in this study: 2021 to 2050 and 2050 to 2080. This means that there was a total of five different climate scenarios, including NCC state based on 2020-2021 modeled data and four climate
altered scenarios (Table 2).

**Table 2: Climate Scenarios Used in the Analysis**

| Climate State | Epoch | Time Period |
|---|---|---|
| NCC | 2021 | 2021 |
| RCP 4.5 | 2050 | 2021 to 2050 |
| RCP 4.5 | 2080 | 2051 to 2080 |
| RCP 8.5 | 2050 | 2021 to 2050 |
| RCP.8.5 | 2080 | 2051 to2080 |

To establish a workflow for rapidly comparing flood hazard exposure from the various JBA flood model estimates, an example exposure dataset was constructed using the open-sourced Microsoft Canadian Building Footprints
(MCBF) dataset (Microsoft, 2019). This dataset contains roughly 11.8 million computer-generated building footprints across Canada using deep learning, computer vision, and Artificial Intelligence techniques rooted in image recognition. Buildings are usually spatially expressed as polygon or point features (Koivumäki et al., 2010), the former representing the buildings shape, and the latter representing a single point usually inside of the structure. The benefit of building footprint data is it includes more information (full shape) and is more likely to capture overlap than a single
point inside of a polygon, which could miss partial flood overlap. Specifically, the MCBF have been used before in flood exposure assessments (Allen et al., 2020; Buchanan et al., 2020; Huang & Wang, 2020; Porter et al., 2023), as it constitutes a readily available source of building location information across entire countries such as Canada and the United States and including more remote areas not mapped by other means.

A spatial computation is needed to determine which buildings are exposed to flooding. Techniques for estimating flood exposure to buildings inherently involve the combination of either a flood extent polygon or flood depth raster file and a vector building dataset. For example, Allen et al. (2020) estimated the number of buildings affected by flooding by computing the intersection of each building footprint with a flood extent polygon. Buchanan et al. (2020) assumed each building polygon was exposed to flooding if it is on land at a lower elevation than a given water height,
and Huang and Wang (2020) estimated exposure as buildings which fall in a floodplain boundary. The approach used in this analysis is an overlap, whereby the hazard value(s) of overlap indicate an exposure, and if multiple depth values are observed, the maximum depth is selected. Intersections of building data with flood extent polygons may not include a depth, only a binary exposure indicator (exposed or not exposed), based on whether a building has any overlap with a flood extent.


MCBF data were downloaded from the Microsoft Github repository, specifically for British Columbia. The data were decompressed, then imported into QGIS in its raw geojson file format. To extract a sample of the MCBF data relevant to the study area, the Clip algorithm was used in QGIS using study boundary file, as shown in Figure 2. This resulted in 103,935 individual building polygons for the Vancouver study area.


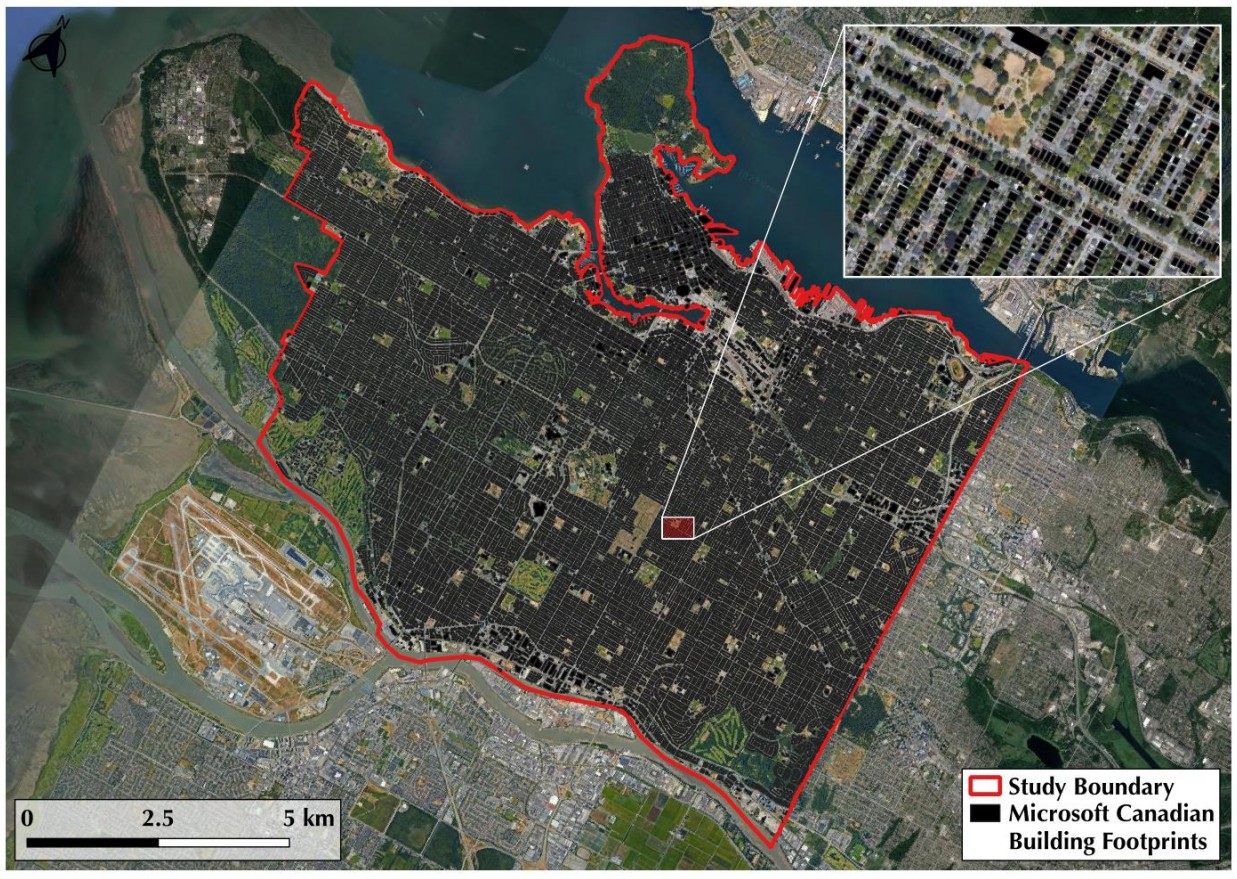

**Figure 2: Study boundary with Microsoft Canadian Building Footprint data (n=103,935). Other map data: Google Satellite Imagery ©2023, TerraMetrics ©2023**

To assess flood exposure across numerous flood hazard scenarios, a Python script was developed which combined the MCBF polygon file with each of the JBA flood scenarios and return periods. Since there were five climate scenarios (CS), three flood mechanisms (FM), and seven return periods (RP), each building was assigned 105 flood depth values (21 per climate scenario):


$$Flood\ hazard\ scenarios = N_{CS}\ x\ N_{FM}\ x\ N_{RP}$$
$$Flood\ hazard\ scenarios = 5\ x\ 3\ x\ 7$$
$$Flood\ hazard\ scenarios = 105$$

To start, the MCBF building polygon dataset was opened as a *geopandas* file in Python. Then, a systematic loop was
implemented which: (1) imported one of the flood hazard files using the *rasterio* package, and (2) computed summary
statistics of the hazard files using the *rasterstats* zonal statistics function for each building. This procedure was
repeated for each of the 105 flood hazard files, summarizing the hazard data at each building polygon. For this
assessment, the maximum depth was chosen as the metric for determining exposure, such that buildings were assigned
the maximum flood hazard depth for each return period intersecting a given building polygon. If any portion of a
building was implicated by flood hazard data, the maximum value of flood depth was assigned.

The practice of using any building overlap is common to flood exposure estimation (Allen et al., 2020; Buchanan et
al., 2020; Huang & Wang, 2020), but more information can be gleaned when also using flood depth at exposure. Since
multiple flood depth measurements could occur for a given building, one must choose which to include for depth-
related assessments based on the purpose of the assessment. Much like Arrighi et al. (2020), zonal statistics were
computed using a series of summary statistics. However, for the purposes of this assessment, the choice of maximum
flood depth at a given building was used to articulate the worst exposure possibility at a given building, which aligns
with other research (Porter et al., 2023). However, some authors have opted to use a combination of maximum and
mean flood depths to factor in outliers of the maximum depth which may be caused by erroneous cells of the terrain
model (Bertsch et al., 2022). For the purposes of this assessment, the maximum depth constitutes the worse of exposure
scenarios for buildings where multiple depth measurements are observed and allows for the estimation of higher in-
building exposure than using other summary metrics. Although there is uncertainty in the selection of a summary
metric, the selection of a maximum depth is reasonable for the purposes of gauging exposure and assessing the effects
of climate change.

For each climate state scenario, flood water depths were associated with each flood mechanism and return period for
each address. To illustrate, Table 3 provides the different flood hazard sampling for the NCC scenario. The same
information provided in Table 3 also applied to the different RCP and time scenarios illustrated in Table 2.

**Table 3: Example CS-FM-RP Level Data at Each Location**

| Item | Climate State | Flood Mechanism | Return Period |
|------|---------------|-----------------|---------------|
| 1-7  | NCC           | Fluvial         | 20            |
|      |               |                 | 50            |
|      |               |                 | 75            |
|      |               |                 | 100           |
|      |               |                 | 200           |
|      |               |                 | 500           |
|      |               |                 | 1500          |
| 8-14 | NCC           | Pluvial         | 20            |

| | | | |
|---|---|---|---|
| | | | 50 |
| | | | 75 |
| | | | 100 |
| | | | 200 |
| | | | 500 |
| | | | 1500 |
| 15-21 | NCC | Storm Surge | 20 |
| | | | 50 |
| | | | 75 |
| | | | 100 |
| | | | 200 |
| | | | 500 |
| | | | 1500 |

The result of this analysis was a building file that contained an associated maximum flood depth at each return period for each flood mechanism and climate state. Despite the numerous return periods available by JBA, we will continue with the remainder of the analysis using the 100-year return period as to focus discussion on differentiating between

sources of flooding (fluvial, pluvial, storm surge), and climate change scenarios. Future works by the authors will leverage multiple return periods to discuss the effects of climate change on different exposures at different return periods.

For this assessment, a flood depth value greater than zero indicated that a given asset was 'exposed' at the

corresponding return period, however, it is important to note that greater depths of water are associated with a higher likelihood of damage or loss. To account for this, three exposure metrics were computed:

1. any building with a flood depth greater than 0 m (any exposure),
2. any building with a flood depth greater than or equal to 0.3 m (moderate exposure),
3. any building with a flood depth greater than or equal to 0.6 m (severe exposure).

These thresholds were selected based on expert consultation and are also based on the 0.3 m or 0.6 m freeboard which is sometimes included in provincial flood maps as an additional margin of safety in the flood elevation (National Research Council of Canada, 2021). These heights approximate 1 foot (0.3048 m) and 2 feet (0.6096 m) and were

selected as general first floor elevation (FFE) possibilities. Although there may be considerable variability in first floor elevation across Vancouver, and subjective variability in the classification of moderate or severe exposure, these values reflect a starting point for the assessment that is informed by Canadian land use information.

These different depths account for differences in first floor elevation and doorstep height across the study area. These

factors, along with other property-level considerations or broader flood defense considerations, may differentiate risk

and whether floodwaters would enter a home and cause damage. From this assessment, an evaluation of individual return period event exposure and the suite of return period exposure was taken to determine where new flood exposure might occur because of climate change. The results of this process are described below.

**4.0 Results**

Overall, there were 103,935 building footprints identified in the study area. Of these, 16,820 (16.2%) were identified as exposed to flooding at the 100-year return period in the NCC condition from *any* flood type or multiple flood types, the latter referring to buildings which may be exposed to more than one flood mechanism at the 100-year return period such as fluvial *and* pluvial flooding. For moderate exposure – that which is greater than or equal to 0.3m – there were 11,050 (10.6%) buildings identified in the study boundary. For severe exposure – that which is greater than or equal

to 0.6m – there were 6,840 (6.6%) buildings identified in the study boundary. Table 4 provides a detailed breakdown of exposure by flood type for the NCC state at the 100-year return period.

**Table 4: Breakdown of FM of Exposure for the NCC state, 100-Year RP Flood Hazard**

| Flood Mechanism(s) | n Exposed (Any) | % of Exposure (Any) | n Exposed (Moderate) | % of Exposure (Moderate) | n Exposed (Severe) | % of Exposure (Severe) |
|---|---|---|---|---|---|---|
| Fluvial | 118 | 0.7 % | 73 | 0.7 % | 34 | 0.5 % |
| Pluvial | 16,252 | 96.6 % | 10,629 | 96.2 % | 6,585 | 96.3 % |
| Storm Surge | 182 | 1.1 % | 214 | 1.9 % | 175 | 2.6 % |
| Fluvial and Pluvial | 71 | 0.4 % | 41 | 0.4 % | 10 | 0.1 % |
| Fluvial and Storm Surge | 124 | 0.7 % | 68 | 0.6 % | 22 | 0.3 % |
| Pluvial and Storm Surge | 45 | 0.3 % | 21 | 0.2 % | 13 | 0.2 % |
| All Three Mechanisms | 28 | 0.2 % | 4 | 0 % | 1 | 0 % |
| TOTAL | 16,820 | 100 % | 11,050 | 100 % | 6,840 | 100 % |

For all types of exposure in Vancouver, the overwhelming majority occurred due to pluvial flooding. Of the 16,820

buildings with any exposure to flooding, 16,252 (96.6%) occurred from pluvial only sourcing. The dominant exposure to pluvial flooding compared to other flood types is common partly because the geographic area is not limited to areas close to rivers or along the coast and because the drainage capacity in urban environments may be insufficient to handle the volumes of rain experienced under current and anticipated climates. The same general distribution of exposure by flood mechanism occurred across any moderate and severe exposure types, with pluvial being the

dominant source; however, there was slightly higher proportions of severe exposure occurring from storm surge compared to estimates when using exposure at any depth.

It is worth noting that pluvial flooding involves a greater degree of uncertainty, in part due to the complexity of incorporating human-generated subsurface drainage infrastructures and waterways. For the purposes of this study,
only drainage capacity assumptions were applied and not specific details on stormwater infrastructure. Though the results indicated that pluvial flooding was the primary driver of exposure in Vancouver, this may not be true in other settings where fluvial or storm surge flooding are the drivers of exposure. Figure 3 shows the same Southlands region of Vancouver with fluvial, pluvial, and storm surge flooding, displaying that though all are implicated, storm surge appears to lead to more severe exposure than fluvial or pluvial flooding. This differentiation is important for
determining flooding that is more likely to cause structural damage due to greater water depths.

When different climate change scenarios are factored in, flood hazard exposure can differ. Table 5, Table 6, and Table 7 provide a detailed breakdown of the different exposures for the 100-year return period under each climate scenario for any exposure, moderate exposure, and severe exposure, respectively. For all three exposure levels, the total number
of building footprints exposed increased in each climate change scenario. For example, using any exposure, the total number of buildings increased from 16,820 to 18,163 (7.98%) for the RCP 8.5 (2050) climate change state. Interestingly, this exposure is similar in magnitude for the RCP 8.5 (2080) scenario, which estimated 18,156 MCBF, a 7.94% increase. Also of interest is that the total number of exposed properties at any depth decreased from RCP 4.5 (2050) to RCP 4.5 (2080), though both showed an estimated increase from the baseline of 16,820. It is generally
accepted that RCP 8.5 is more reflective of the projected climate state than RCP 4.5. The same general distributions of flood exposure occurred by flood type, with pluvial-sourced flooding the dominant flood type leading to exposure in the study area of Vancouver.

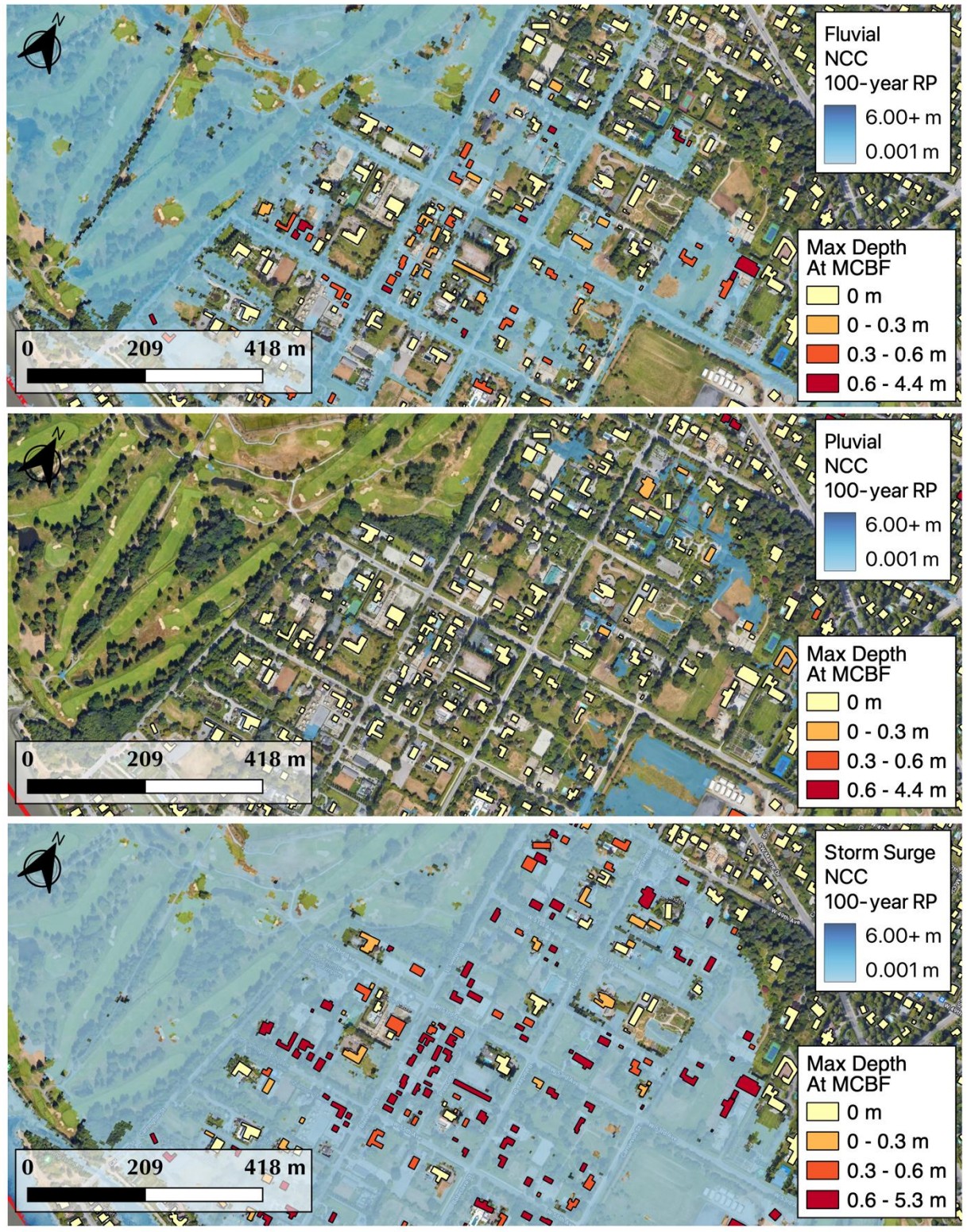

Figure 3: Flood exposure from fluvial, pluvial, and storm surge flooding for any exposure (> 0m), moderate exposure (>= 0.3m), and severe exposure (>=0.6m) in the Southlands of Vancouver (top and bottom) and northeastern reaches of the Arbutus Ridge Neighbourhood in Vancouver (middle). Other map data: Google Satellite Imagery ©2023, CNES / Airbus, Maxar Technologies ©2023

*Table 5: Breakdown of Any Exposure by FM for the Various Climate States at the 100-Year RP Flood Hazard*

| Flood Mechanism(s) | n Exposed (NCC) | % of Exposure (NCC) | n Exposed (RCP 4.5, 2050) | % of Exposure (RCP 4.5, 2050) | n Exposed (RCP 8.5, 2050) | % of Exposure (RCP 8.5, 2050) | n Exposed (RCP 4.5, 2080) | % of Exposure (RCP 4.5, 2080) | n Exposed (RCP 8.5, 2080) | % of Exposure (RCP 8.5, 2080) |
|---|---|---|---|---|---|---|---|---|---|---|
| Fluvial | 118 | 0.7 % | 133 | 0.8 % | 151 | 0.8 % | 124 | 0.7 % | 143 | 0.8 % |
| Pluvial | 16,252 | 96.6 % | 16,767 | 96.2 % | 17,451 | 96.1 % | 16,543 | 95.8 % | 17,390 | 95.8 % |
| Storm Surge | 182 | 1.1 % | 212 | 1.2 % | 205 | 1.1 % | 235 | 1.4 % | 235 | 1.3 % |
| Fluvial and Pluvial | 71 | 0.4 % | 47 | 0.3 % | 76 | 0.4 % | 44 | 0.3 % | 57 | 0.3 % |
| Fluvial and Storm Surge | 124 | 0.7 % | 142 | 0.8 % | 160 | 0.9 % | 142 | 0.8 % | 171 | 0.9 % |
| Pluvial and Storm Surge | 45 | 0.3 % | 97 | 0.6 % | 87 | 0.5 % | 124 | 0.7 % | 113 | 0.6 % |
| All Three Mechanisms | 28 | 0.2 % | 39 | 0.2 % | 33 | 0.2 % | 50 | 0.3 % | 47 | 0.3 % |
| **TOTAL** | **16,820** | **100 %** | **17,437** | **100 %** | **18,163** | **100 %** | **17,262** | **100 %** | **18,156** | **100 %** |
| **TOTAL Δ from NCC** | • | • | **617** | **3.67%** | **1,343** | **7.98%** | **442** | **2.63%** | **1,336** | **7.94%** |

*Table 6: Breakdown of Moderate Exposure by FM for the Various Climate States at the 100-Year RP Flood Hazard*

| Flood Mechanism(s) | n Exposed (NCC) | % of Exposure (NCC) | n Exposed (RCP 4.5, 2050) | % of Exposure (RCP 4.5, 2050) | n Exposed (RCP 8.5, 2050) | % of Exposure (RCP 8.5, 2050) | n Exposed (RCP 4.5, 2080) | % of Exposure (RCP 4.5, 2080) | n Exposed (RCP 8.5, 2080) | % of Exposure (RCP 8.5, 2080) |
|---|---|---|---|---|---|---|---|---|---|---|
| Fluvial | 73 | 0.7 % | 84 | 0.7 % | 104 | 0.9 % | 77 | 0.7 % | 97 | 0.8 % |
| Pluvial | 10,629 | 96.2 % | 10,960 | 95.7 % | 11,354 | 95.6 % | 10,815 | 95.3 % | 11,324 | 95 % |
| Storm Surge | 214 | 1.9 % | 248 | 2.2 % | 237 | 2 % | 283 | 2.5 % | 287 | 2.4 % |
| Fluvial and Pluvial | 41 | 0.4 % | 26 | 0.2 % | 44 | 0.4 % | 22 | 0.2 % | 36 | 0.3 % |
| Fluvial and Storm Surge | 68 | 0.6 % | 76 | 0.7 % | 91 | 0.8 % | 77 | 0.7 % | 96 | 0.8 % |
| Pluvial and Storm Surge | 21 | 0.2 % | 45 | 0.4 % | 43 | 0.4 % | 59 | 0.5 % | 57 | 0.5 % |
| All Three Mechanisms | 4 | 0 % | 13 | 0.1 % | 9 | 0.1 % | 20 | 0.2 % | 18 | 0.2 % |
| **TOTAL** | **11,050** | **100 %** | **11,452** | **100 %** | **11,882** | **100 %** | **11,353** | **100 %** | **11,915** | **100 %** |
| **TOTAL Δ from NCC** | • | • | **402** | **3.64%** | **832** | **7.53%** | **303** | **2.74%** | **865** | **7.83%** |

*Table 7: Breakdown of Severe Exposure by FM for the Various Climate States at the 100-Year RP Flood Hazard*

| Flood Mechanism(s) | n Exposed (NCC) | % of Exposure (NCC) | n Exposed (RCP 4.5, 2050) | % of Exposure (RCP 4.5, 2050) | n Exposed (RCP 8.5, 2050) | % of Exposure (RCP 8.5, 2050) | n Exposed (RCP 4.5, 2080) | % of Exposure (RCP 4.5, 2080) | n Exposed (RCP 8.5, 2080) | % of Exposure (RCP 8.5, 2080) |
|---|---|---|---|---|---|---|---|---|---|---|
| Fluvial | 34 | 0.5 % | 35 | 0.5 % | 49 | 0.7 % | 33 | 0.5 % | 45 | 0.6 % |
| Pluvial | 6,585 | 96.3 % | 6,837 | 95.4 % | 7,133 | 95.3 % | 6,723 | 94.8 % | 7,114 | 94.6 % |
| Storm Surge | 175 | 2.6 % | 237 | 3.3 % | 235 | 3.1 % | 269 | 3.8 % | 283 | 3.8 % |
| Fluvial and Pluvial | 10 | 0.1 % | 7 | 0.1 % | 16 | 0.2 % | 6 | 0.1 % | 14 | 0.2 % |
| Fluvial and Storm Surge | 22 | 0.3 % | 26 | 0.4 % | 28 | 0.4 % | 26 | 0.4 % | 31 | 0.4 % |
| Pluvial and Storm Surge | 13 | 0.2 % | 19 | 0.3 % | 19 | 0.3 % | 30 | 0.4 % | 29 | 0.4 % |
| All Three Mechanisms | 1 | 0 % | 3 | 0 % | 1 | 0 % | 5 | 0.1 % | 4 | 0.1 % |
| **TOTAL** | **6,840** | **100 %** | **7,164** | **100 %** | **7,481** | **100 %** | **7,092** | **100 %** | **7,520** | **100 %** |
| **TOTAL Δ from NCC** | • | • | 324 | 4.74% | 641 | 9.37% | 252 | 3.68% | 680 | 9.94% |

Of the 16,820 MCBF considered exposed in the baseline climate scenario (Table 5), 11,050 (65.7%) were considered to have moderate exposure (Table 6) and 6,840 (40.7%) were considered to have severe exposure (Table 7). Though pluvial flooding is the dominant source of all three exposure levels, storm surge has an increasing proportion of severe exposure in all climate change scenarios (Table 7). Although the total exposure increases for all climate scenarios, the effect of climate change seems

more pronounced for severe exposure. Specifically, severe exposure increased for RCP 8.5 2050 and RCP 8.5 2080 between 9.3% and 10%, while any exposure and moderate exposure showed increases between 7.5% and 8.0% (Tables 5-7).

For a more detailed breakdown of the change in exposure associated with climate change projections, asset exposure was disaggregated into three general categories: (1) continued exposure, (2) new exposure, and (3) former exposure. Continued

exposure refers to assets that were considered exposed to flooding at a given return period both in a non-climate change state and given altered climate change state, and is further broken down into three sub-categories: (A) lessened exposure, (B) similar exposure, and (C) worsened exposure. New exposure refers to the assets that were not considered exposed to flooding at a given return period but are exposed under an altered climate state. Former exposure refers to the assets that were considered exposed to flooding at a given return period but were not exposed under an altered climate state. To compute each sub-category

of continued exposure, the flood depths at each building were rounded to the nearest millimetre as to avoid exceedingly small changes from being classified as worsened or lessened exposures when differences are negligible.

Of the 16,820 buildings that were considered exposed to any 100-year flood type and depth (i.e., fluvial, pluvial, and storm surge) under the NCC state, 16,806 continued to be exposed in the RCP 8.5 2080 climate state at any depth, while 14 were no

longer considered exposed. Of the continued exposure (n=16,806), 16,051 (95.5%) worsened in the RCP 8.5 2080 climate state, 640 (3.8%) had similar flood depths, and 115 (0.7%) lessened in flood depth. Additionally, there were 1,350 new assets exposed to flooding due to climate change. This is summarized in Table 8. The result is a total estimated exposure of 18,156 buildings for the RCP 8.5 2080 climate state at any depth. An overview of this comparison is provided in Figure 4.

*Table 8: Exposure breakdown between the baseline NCC state and the RCP 8.5 2080 climate state for the 100-year return period using all flood mechanisms*

| Exposure Breakdown | n |
| --- | --- |
| **Continued Exposure** | **16,806** |
| Lessened Exposure | 115 |
| Similar Exposure | 640 |
| Worsened Exposure | 16,051 |

| Exposure Breakdown | n |
| --- | --- |
| New Exposure | 1,350 |
| No Exposure | 85,765 |
| Former Exposure | 14 |

For the 16,806 buildings classified as having 'continued' exposure both at the NCC and RCP 8.5 2080 climate states, we observed differences in the proportion of exposure that is considered moderate and severe. Of the 16,806 buildings, 11,044 (65.7%) were considered moderate at NCC conditions whereas 11,536 (68.6%) were considered moderate at RCP 8.5 2080 conditions, a 4.5% (n = 492) increase in the moderately exposed buildings. Severe exposure increased as well. Of the 16,806 buildings, 6,837 (40.7%) were considered severely exposed at the NCC conditions, while 7,320 (43.6%) were considered severely exposed at the RCP 8.5 2080 conditions. This reflects a 7% (n = 483) increase in the severely exposed buildings. The exposed properties were distributed all over the study site, which is to be expected from the widespread pluvial flood hazard estimation.

The overwhelming majority of these assets continued to be exposed under the RCP 8.5 2080 climate state and, due to the diffuse nature of pluvial flooding, the new exposure was scattered throughout the study site. As depicted in Figure 4, in some cases we noted larger amounts of water leading to further exposure on the periphery of regions affected under the NCC state (inset map of Figure 4). Such instances would be explained by more water occurring from flooding in the same areas exposed in the baseline non-climate change state, leading to an expansion of the flood extent and greater depths of water at previously exposed buildings. Further, of the 'continued' exposure, the average change in flood depth at each building was a roughly 0.04 m increase (-0.62 m to 1.53m range), although some buildings did continue to be exposed in the climate change condition but at a lesser flood depth (n=115).

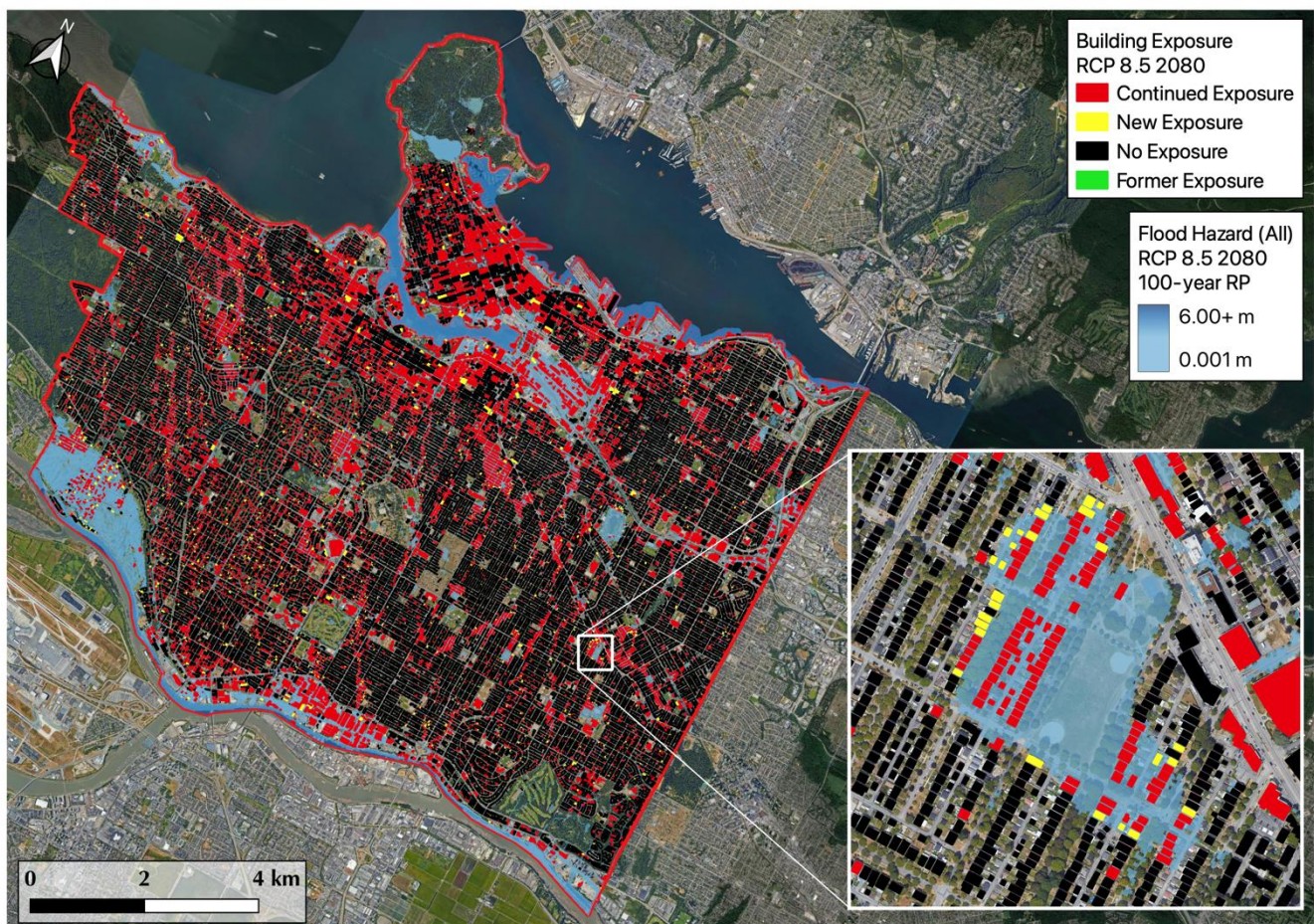

**Figure 4: Overview of Continued, New, and Former Exposure in Vancouver using the RCP 8.5 2080 Climate State at the 100-year return period. Other map data: Google Satellite Imagery ©2023, TerraMetrics ©2023 CNES / Airbus, Maxar Technologies ©2023**

Generally, we expected that most properties exposed in a NCC state would also be exposed in a climate-altered state, largely due to the expectation that all major types of flooding, including fluvial (riverine), pluvial (rainfall) and storm surge (coastal) will intensify as the climate changes (Alfieri et al., 2016; Arnell & Gosling, 2016; Hirabayashi et al., 2021; IPCC, 2019; Muis et al., 2016; Winsemius et al., 2016) and the similar assumptions embedded into the JBA climate change modules involving precipitation, temperature, and run-off. A few buildings (n = 14) were determined to be exposed at the baseline non-climate change state that were classified as non-exposed at RCP 8.5 2080. For example, Figure 5 illustrates two of the 14 assets classified as 'former' exposure, which were considered exposed in the NCC state but not under the RCP 8.5 2080 climate state at the 100-year return period. Ultimately, these were investigated individually and determined to be all pluvial-sourced flooding at shallow flood depths and almost always occurred in small pockets of disjoint pools of water. These very small artifacts in map output (e.g., where flooding in the future map was not present in the non-climate change state) reflect the complexity in

the modelling process, and for example, the way the rounding of parameters can propagate through to the end map slightly

differently.

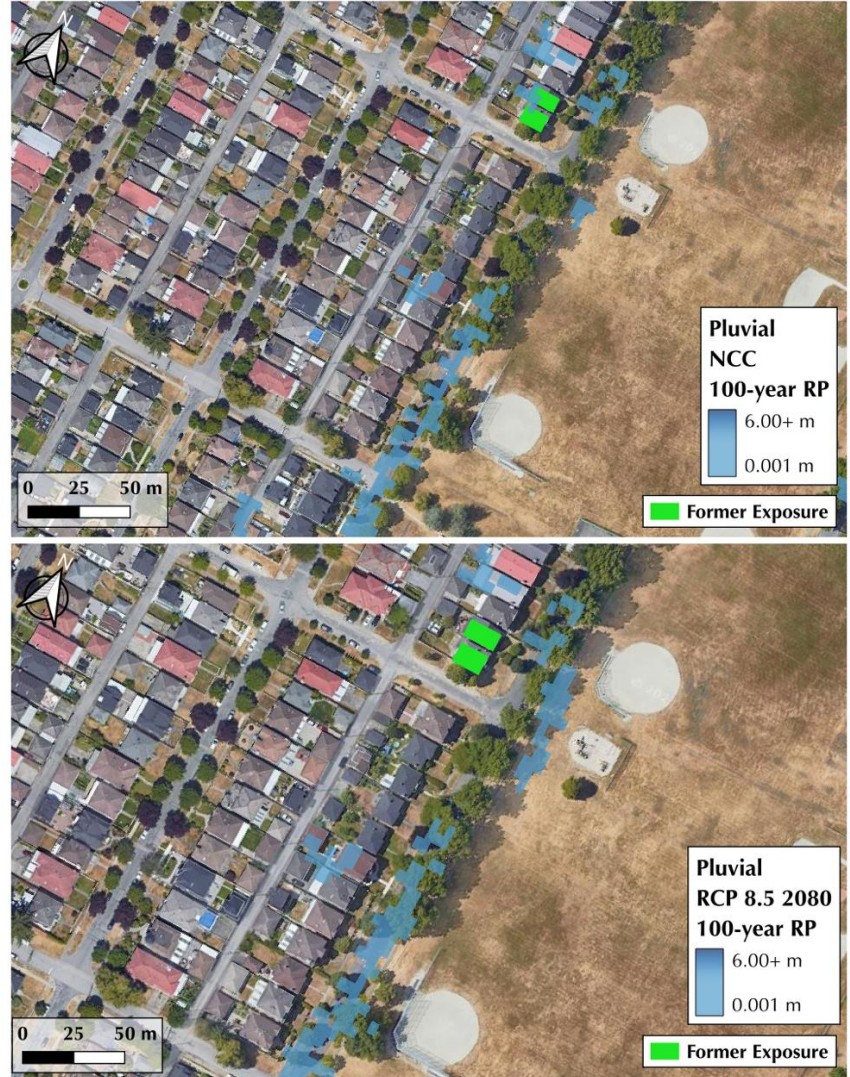

**Figure 5: Example of Two Assets Considered Exposed at the NCC State and not at RCP 8.5 2080 Climate State. Other map data: Google Satellite Imagery ©2023, CNES / Airbus, Maxar Technologies ©2023**

**5.0 Discussion**

The Vancouver case study demonstrates the applicability and granularity of the flood hazard mapping tool for identifying local

flood exposure under a changing climate. This simplified flood hazard mapping approach was able to capture changes to flood

exposure based on different climate states, flood types, and return periods. Interestingly, most of the engineered flood hazard

mapping in Canada pertains to fluvial flooding (MMM Group Limited, 2014), meaning pluvial and storm surge inclusions could reflect an uncaptured source of flood risk in Vancouver by stakeholders relying on publicly available engineered maps

only. Given that the results of this study indicate that pluvial flooding is the driving mechanism for anticipated flood exposure in the Vancouver area, excluding this in flood mapping could result in harmful consequences on the accuracy and totality of flood hazard maps. Moreover, its absence from planning could lead to decision-making that puts vulnerable populations at higher risk to disasters by enabling further development decisions in high-risk zones, an issue that is a significant driver of current disaster risk in Canada and elsewhere.


More broadly, the higher resolution data (5-metre) constitutes an improvement from 30+ metre resolution models being produced nationally, although local engineered mapping is typically around 2-metre resolution. By modeling climate change, this type of exposure assessment enables mapping of changes to flood exposure for future climate states. Other benefits of this approach include its capacity to differentiate flood exposure based on the source of flooding, its scalability, and its ability to

provide a generalized analysis for planners and policymakers. This approach is deemed scalable as it can be replicated in any other location in Canada, with researchers needing to only substitute the flood hazard and exposure data of interest into the established code, and the analysis can be re-run in the new setting. Although the availability of 5m resolution and climate change data is not universally available at present, JBA offers 30m non-climate change coverage in all areas across Canada. Although there is greater uncertainty in the pluvial flood estimation, largely because nearly all regulatory flood mapping to

compare and calibrate against is fluvial and city-specific engineered drainage systems are not typically available, it constitutes a largely unmapped source of risk that can become increasingly important in urban environments.

The use of higher resolution hazard data, climate change conditions and multiple flood mechanisms in this analysis improves upon mapping techniques that are often low resolution, lack climate change considerations, and are highly complex to develop

(Cea & Costabile, 2022; Costabile et al., 2015; de Moel et al., 2009; Mudashiru et al., 2021; Teng et al., 2017). Moreover, this mapping approach demonstrates the relevance of physically-based modeling as a modern approach to flood hazard mapping that is less costly and onerous to produce than conventional approaches. This may be especially appealing to planners and engineers who can now more effectively argue that some communities should have policies encouraging property-level flood protections.


Although the model and case study demonstrate a general and easy-to-use approach to local flood hazard mapping, they have some limitations. First, the analysis does not distinguish between building types. When reviewing Figure 4 more closely, some structures were not identified as buildings based on the exposure dataset. Most of these structures are assumed to be sheds, but their exposure is still relevant. While this represents a limitation of the MCBF data, the use of this data in this model is a

strength as well. This limitation is a product of the MCBF and its coding. In many parts of the world, modeling flood exposure

is limited by poor or even non-existent exposure datasets, such that assumptions must be made by scientists regarding population density. Here, the produced flood hazard maps show that despite some limitations within the datasets used, the overall quality of the model remains high.

Second, this model used a constant exposure dataset based on recent building data availability (2019). Conditions in 2050 and 2080 would also include more buildings and more exposure, which can constitute another considerable source of additional flood risk. It is anticipated that future development will lead to higher risks under future climate conditions because of the anticipated added density and number of buildings that will be exposed to flooding. For this analysis, exposure was held constant while flood hazard was modified to different climate change scenarios. Future research could consider the influence
of new development on additional flood risk.

Third, this method for flood hazard mapping identified buildings using an estimated polygon from the source data, but it does not account for building information such as building type (residential, commercial, industrial, etc.). This approach makes it difficult to differentiate flood exposure by land use, classification, or for other relevant characteristics. Though there are
inherent inaccuracies and a recency problem to the dataset – which was released in 2019 – the general flood exposure approach was the focus of this paper.

Finally, the produced flood hazard maps using the model presented here are considered by industry standards to be a higher resolution than most other models, given that most maps are produced at a 30-metre or smaller resolution. Local flood models
that use physical methods to identify flood exposure could further improve the precision of these results by presenting flood exposure at an even finer scale (e.g., at a 1-metre or 2-metre resolution). This becomes computationally challenging, especially if the spatial extent of the area grows from a single city to a provincial or even national scale, to offer consistency. Further, comparisons and validation of the flood hazard data against local engineered mapping or past flood events could reveal the hazard accuracy to known events or to on-the-ground results in Vancouver. The strengths of this approach, however, are a
reasonable trade-off since this efficient flood hazard mapping approach nevertheless produces visual outputs that would be valuable for future land use planning, emergency management, policymaking, and risk awareness. Additionally, the approach may be appealing to smaller municipalities that lack resources to pursue higher-resolution modeling.

Validation of the model results is needed to rectify some of these issues. For example, a manual inspection of structures
throughout the study area and comparing the results of this model against others that have higher and lower resolutions would allow us to monitor the overall accuracy of the findings. Like many other 2D hydrologic models, the JBA model makes spatial and temporal assumptions on the heterogeneous properties of the local environment (see Abbaszadeh et al., 2022; Anees et al., 2017). These assumptions are a source of uncertainty that need to be tested and calibrated by investigating the effects of model

inputs (Willis et al., 2019). In the absence of any validating experiments, these maps should be viewed through a cautionary lens. Moreover, this model identifies changes to exposure due to climate change but does not consider changing socioeconomic characteristics of the area or economic consequences of flooding, nor does it include existing drainage or stormwater storage capacity, which may impact the results. These are areas requiring further research.

**6.0 Conclusion**

The acceleration of flood risk caused by climate change and expanding development in flood-prone areas requires local flood hazard maps that will enable governments, non-governmental organizations, and others to plan and implement interventions that will protect assets and populations. However, flood hazard maps often fail to account for climate change, are developed through highly technical methodologies, depict exposure to only one source of flooding, and have a course resolution. Leading scholarship has suggested that simplified mapping approaches are needed to overcome these weaknesses while permitting practical use by non-experts.

This paper presented a simple modeling approach to produce local flood hazard maps using a moderate 5-metre resolution based on JBA Risk Management's 5-metre Baseline and Climate Change Flood Map Data for Canada. In using these data, we demonstrated an empirical approach to local flood hazard mapping that produces generalizable flood exposure information. The approach can model changes to flood exposure based on the flood type, climate change projections, and return periods. This is novel to flood hazard mapping because of its scalability and its ability to capture climate change and pluvial flood exposure. This is the first study that maps Vancouver's current exposure to fluvial, pluvial and coastal flood at a 5-metre resolution and based on climate change information. While other models, like First Street Foundation, FLO-2D and Fathom, do provide physically-based solutions to mapping flood exposure across large geographic areas, including at a fairly high resolution, no such models exist in Canada. This represents a significant cap in geographic coverage for a country that experiences a high volume of floods annually. Moreover, as the most widely used model in the Canadian insurance market, this study demonstrates the utility of the JBA model at providing comprehensive coverage for fluvial, pluvial, and coastal flooding. While certain limitations do exist – including the inability to differentiate exposure based on building type (e.g., residential versus commercial) – the approach enables local planners, policymakers, and other stakeholders to pursue flood risk management strategies.

Further research is needed to validate the findings of the current model and its replicability in other jurisdictions. Comparing the resulting local flood hazard map with maps produced using other methodologies – including physical and empirical models – and against historical flood events would allow us to validate the findings even further. Validating climate change hazard

estimation remains a challenge due to the uncertainty surrounding climate change effects; however, this model constitutes a
step forward in the use of forward-looking flood hazard and exposure estimation.

**Data Availability**

As per the policy on data sharing, access to data used in this research can be made available upon reasonable request to JBA Risk Management. Access to the MCBF data can be found on GitHub (GitHub - microsoft/CanadianBuildingFootprints: Computer generated building footprints for Canada).

**Author Contributions**

Connor Darlington developed the research methodology and carried out the data analysis. Jonathan Raikes prepared the manuscript with contributions from all co-authors.

**Competing Interests**

The authors declare that they have no conflict of interest.

**Acknowledgements**

The authors thank JBA Risk Management Limited for providing the data necessary to produce this analysis. This research was funded by the Social Sciences and Humanities Research Council of Canada (grant #435-2018-0377).

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
