# Peer review of "Mapping current and future flood exposure using a 5-metre flood model and climate change projections"

_Natural Hazards and Earth System Sciences, 2023_

## Referee Comment (RC1)

Peer-review of NHESS article: *Mapping current and future flood exposure using a 5-metre flood model and climate change projections*

**Questions for consideration:**

1. Does the paper address relevant scientific and/or technical questions within the scope of NHESS? Yes
2. Does the paper present new data and/or novel concepts, ideas, tools, methods or results? Uncertain - reasoning explained below
3. Are these up to international standards? Yes
4. Are the scientific methods and assumptions valid and outlined clearly? No - Need more details on the modeling that produced the dataset used
5. Are the results sufficient to support the interpretations and the conclusions? Unclear, explained below
6. Does the author reach substantial conclusions? Unclear, explained below
7. Is the description of the data used, the methods used, the experiments and calculations made, and the results obtained sufficiently complete and accurate to allow their reproduction by fellow scientists (traceability of results)? Again, would like more details on the modeling methods. But the methods the authors used were clear and reproducible.
8. Does the title clearly and unambiguously reflect the contents of the paper? Yes
9. Does the abstract provide a concise, complete and unambiguous summary of the work done and the results obtained? Yes
10. Are the title and the abstract pertinent, and easy to understand to a wide and diversified audience? Yes
11. Are mathematical formulae, symbols, abbreviations and units correctly defined and used? If the formulae, symbols or abbreviations are numerous, are there tables or appendixes listing them? Yes
12. Is the size, quality and readability of each figure adequate to the type and quantity of data presented? Yes
13. Does the author give proper credit to previous and/or related work, and does he/she indicate clearly his/her own contribution? Yes
14. Are the number and quality of the references appropriate? Would like to see a more detailed review of more papers - explained further below
15. Are the references accessible by fellow scientists? Yes
16. Is the overall presentation well structured, clear and easy to understand by a wide and general audience? Yes
17. Is the length of the paper adequate, too long or too short? Appropriate length
18. Is there any part of the paper (title, abstract, main text, formulae, symbols, figures and their captions, tables, list of references, appendixes) that needs to be clarified, reduced, added, combined, or eliminated? Figures 1 and 2 could be merged
19. Is the technical language precise and understandable by fellow scientists? Yes

20. Is the English language of good quality, fluent, simple and easy to read and understand by a wide and diversified audience? Yes
21. Is the amount and quality of supplementary material (if any) appropriate? N/A

**General comments:**

- Novelty uncertain - In the review, different modeling approaches are described (e.g., physical, physically-based, and empirical), but there is no review on the current state of physically-based modeling which is the modeling method used. A review on physically-based approaches and their applications is needed to demonstrate the need for and novelty of this work. Similar datasets exist from First Street Foundation and Fathom. Comparing this dataset to existing ones can provide further context of this work.
- Modeling details - The description of the modeling methods was very general. More details are needed to understand how the dataset used was produced, especially for pluvial modeling since pluvial was the driving mechanism in the results.
- Include more return periods in analysis - There were several return periods listed that the data was available for, but results were only reported for the 100-year event. Consider comparing the driving flood mechanism across multiple return periods as this could yield interesting and innovative results.
- Paper is very well-written and organized. It is clear and easy to read.

**Line-specific comments:**

Pg. 2, line 35 - The IPCC citation is not included in the reference list and not in alphabetical order

Pg. 2, line 40 - change anticipating to anticipate

Pg. 4, line 106 - Section is not numbered

Pg. 4, line 142 - Is citation reference to the 2022a or 2022b reference?

Pg. 5, line 154 - The review of Canadian flow mapping throughput history and different initiatives provides nice context.

Figure 1 - Consider condensing Figures 1 and 2. Could add a fourth panel to Figure 2 that shows the inundation including all flood mechanisms for the 100-yr event and eliminating Figure 1.

Pg. 6, line 176 - use of the 'coverage observed' - this could be confusing for readers since this isn't referring to a historical, observed event. Consider changing the language to 'coverage modeled' or 'observed from modeling.'

Pg. 6, line 177 - The use of low to high scale. Figure 2 includes depth ranges instead of qualitative scale. Could the flood depths scale be used for Figure 1?

Pg. 6, line 184 - Figure 2 captions says its mapping the 100-yr event not the 1,500-year RP.

Pg. 8, line 198 - "infiltration coefficient" - I would like to see more details on how this was calculated and applied. Was it assumed to be the same across the study site, or did it vary based on land use or % imperviousness or some other method?

Pg. 8, line 210 - Include more detail on these "change factors" - how are they developed and applied.

Pg. 8, line 216 - Again, more detail on "change factors." Were the change factors for the hydrographs calculated differently than the change factors for the rainfall?

Page 12, line 283 - make "elevation" plural

Page 12, line 285 - Was there consideration of making additional thresholds for measuring the severity of building exposure? Could include an additional threshold equivalent to the height of a first floor since the flood depths exceed 6 meters in some scenarios.

Page 12, line 292 - "combination of flood types" - Does this mean a building was flooded by both or all three flood types? Add a brief clarification as to how this was determined.

Page 12, line 292 - Change "combination" to "multiple" because the modeling considered the flood mechanism separately, and did not implement compound modeling where the mechanisms are included simultaneously in the model. Combination might imply flood mechanisms occurring at the same time and be misleading.

Page 13, line 300 - Results show pluvial is the driving mechanism of flood exposure. Why do you suppose that is? Consider adding this to the discussion.

Page 13, line 306 - Mention the role of "infrastructures and waterways." Does the model consider stormwater infrastructure?

Figure 4 - Is the middle map of a different location? Why not use the same location for the pluvial flood mechanism as well?

Pg. 19, line 32 - Word choice of the word "greater." Consider changing the first instance to "larger amounts of water" and the second instance to "further exposure" to add clarity.

Figure 5 - Consider splitting the classification of "continued exposure" into two groups: "continued exposure" and "worsened exposure" based on a building going from moderate to

severe exposure. This could add more information to the figure to show not only where new exposure occurred but also where it was made worse by climate change.

Page 20, line 41 - What model assumptions are being referred to? Be specific.
Page 20, line 43 - "two of the three assets." The sentence before mentions 14 buildings, so not sure where three assets are coming from.

Page 20, line 46 - the small pockets of disjointed water causing changes. Perhaps this is an opportune place to discuss the role of model uncertainty on the output since the only input that was changed was increasing rainfalls based on climate change.

Page 21, lines 54-56 - can add that pluvial flooding was the driving mechanism, so excluding this in flood mapping could result in harmful consequences

Page 21, line 61 - Provide more detail on the approaches' scalability.

Page 21, line 62 - Why is there greater uncertainty in the pluvial flood estimation?

Page 22, line 83 - make 'building' plural

Page 22, line 84 - Add clarity to wording. Buildings are not a source of exposure, but an increase in development leads to a greater density and number of buildings that can be exposed to flooding.

Page 22, line 85 - Again, buildings are not a source of flooding, but they increase the number of assets that can be exposed.

Page 22, lines 95-96 - Sentence is unclear.

Page 22, line 98: Validation is discussed, but what about the roles of uncertainty of and sensitivity analysis on the dataset and the results?

Page 24, line 161 - Reference is in the reference list but not cited in the manuscript

Page 27, line 236 - Reference is in the reference list but not cited in the manuscript

---

## Author Response (AR1)

We are pleased to respond to the helpful and constructive comments of the two anonymous referees, which were posted on the NHESS Discussion page on July 31, 2023 and September 11, 2023, respectively. The referees' comments and our responses are presented below.

**Reviewer 1**

**1. Novelty uncertain - In the review, different modeling approaches are described (e.g., physical, physically-based, and empirical), but there is no review on the current state of physically-based modeling which is the modeling method used. A review on physically-based approaches and their applications is needed to demonstrate the need for and novelty of this work. Similar datasets exist from First Street Foundation and Fathom. Comparing this dataset to existing ones can provide further context of this work.**

RESPONSE: We have expanded the literature review to include a more fulsome discussion of the state of physically-based modeling, which further contextualizes the study. We also reflect on this state in the discussion and conclusion sections to make the novelty of the presented method more apparent. Recognizing that other models do exist (e.g., First Street Foundation, Fathom, FLO-2D), none of them cover Canadian cities. This is the first comprehensive model that effectively maps fluvial, pluvial, and storm surge flood exposure in Vancouver, Canada at a 5-metre resolution and that estimates future flood exposure based on climate change projections.

**2. Modeling details - The description of the modeling methods was very general. More details are needed to understand how the dataset used was produced, especially for pluvial modeling since pluvial was the driving mechanism in the results.**

RESPONSE: We have revised the description of the modeling method in Section 3.2 to include more details on how the dataset was produced.

**3. Include more return periods in analysis - There were several return periods listed that the data was available for, but results were only reported for the 100-year event. Consider comparing the driving flood mechanism across multiple return periods as this could yield interesting and innovative results.**

RESPONSE: We agree with the reviewer's recommendation that multiple return periods will be interesting to examine and aim to do so in subsequent research. For this research, however, we felt that adding more return periods may detract from the focus on different sources of flooding and the use of climate change scenarios. The more variability we added, the longer the paper would be, and the less it would delve into each area of flood risk. As we are currently at the page limit, we felt it best to discuss return period variability in a future work. We have added some material explaining this after Table 3.

**4. Line-specific comments.**

Pg. 2, line 35 - The IPCC citation is not included in the reference list and not in alphabetical order
RESPONSE: The IPCC in-text citation has been added to the reference list and the citations were re-listed in alphabetical order.

Pg. 2, line 40 - change anticipating to anticipate

RESPONSE: "Anticipating" was changed to "anticipate".

Pg. 4, line 106 - Section is not numbered
RESPONSE: Section number was added.

Pg. 4, line 142 - Is citation reference to the 2022a or 2022b reference?
RESPONSE: We updated the citation to reference Natural Resources Canada, 2022a.

Pg. 5, line 154 - The review of Canadian flow mapping throughput history and different initiatives provides nice context.
RESPONSE: Thank you.

Figure 1 - Consider condensing Figures 1 and 2. Could add a fourth panel to Figure 2 that shows the inundation including all flood mechanisms for the 100-yr event and eliminating Figure 1.
RESPONSE: The recommended change has been implemented. Figures 1 and 2 have been combined into a single figure, all harmonized to the 100-year return period.

Pg. 6, line 176 - use of the 'coverage observed' - this could be confusing for readers since this isn't referring to a historical, observed event. Consider changing the language to 'coverage modeled' or 'observed from modeling.'
RESPONSE: The language was changed to "observed from modeling".

Pg. 6, line 177 - The use of low to high scale. Figure 2 includes depth ranges instead of qualitative scale. Could the flood depths scale be used for Figure 1?
RESPONSE: This change has been reflected with the comment listed above for Figure 1, which recommended condensing Figures 1 and 2. Now, all of the maps use depths instead of the relative terms 'low' and 'high'.

Pg. 6, line 184 - Figure 2 captions says its mapping the 100-yr event not the 1,500-year RP.
RESPONSE: The reference to Figure 2 was amended to reflect the 100-yr event.

Pg. 8, line 198 - "infiltration coefficient" - I would like to see more details on how this was calculated and applied. Was it assumed to be the same across the study site, or did it vary based on land use or % imperviousness or some other method?
RESPONSE: We added some clarity in Section 3.2 about how the infiltration coefficient is determined and applied. JBA uses different coefficients in its modelling to represent different land-use (rural, semi-urban, urban and so on). However, for this study, the same consistent infiltration coefficient was used across the whole urban area. We recognize that this is a limitation because the true infiltration capacity at any given location will vary, but very detailed information on drainage isn't always available, or more importantly, is not always reliable. Hence a general assumption was applied.

Pg. 8, line 210 - Include more detail on these "change factors" - how are they developed and applied.
RESPONSE: In the revised manuscript, we have expanded the description of the method used to calculate climate change-adjusted flood extents.

Pg. 8, line 216 - Again, more detail on "change factors." Were the change factors for the hydrographs calculated differently than the change factors for the rainfall?

RESPONSE: As noted, the revised manuscript explains how climate change-adjusted flood extents were calculated for pluvial, fluvial and coastal flooding.

Page 12, line 283 - make "elevation" plural
RESPONSE: We made elevation plural.

Page 12, line 285 - Was there consideration of making additional thresholds for measuring the severity of building exposure? Could include an additional threshold equivalent to the height of a first floor since the flood depths exceed 6 meters in some scenarios.
RESPONSE: There was consideration for making additional thresholds for measuring the severity of building exposure, but the existing thresholds were deemed sufficient for primary purposes of the paper: providing a mechanism for identifying possible present and future flood risk at a higher resolution than is available at Canada-wide coverage. We will consider doing a deeper-dive assessment for buildings with, say, greater than 1m of water (3.4% of the buildings) at the non-climate change 100-year return period state). Further, only 16 buildings have greater than 4m of depth at the non-climate change 100-year return period state. Though, such thresholds would be arbitrary and do not add considerably more value to the articles main purpose. We will certainly consider this in future works.

Page 12, line 292 - "combination of flood types" - Does this mean a building was flooded by both or all three flood types? Add a brief clarification as to how this was determined.
RESPONSE: "Combination of flood types" was changed to "multiple flood types" for clarity, and the following was further added: "… the latter referring to buildings which may be exposed to more than one flood mechanism at the 100-year return period such as fluvial *and* pluvial flooding."

Page 12, line 292 - Change "combination" to "multiple" because the modeling considered the flood mechanism separately, and did not implement compound modeling where the mechanisms are included simultaneously in the model. Combination might imply flood mechanisms occurring at the same time and be misleading.
RESPONSE: We changed "combination" to "multiple".

Page 13, line 300 - Results show pluvial is the driving mechanism of flood exposure. Why do you suppose that is? Consider adding this to the discussion.
RESPONSE: We have added an explanation about why pluvial is the driving mechanism of flood exposure, noting that it is common for pluvial flooding to be the dominant flood type partly because its geographic coverage is not limited to areas close to rivers or the coast and because the drainage capacity in urban environments may be insufficient to handle the volumes of rain experienced under current and anticipated climates.

Page 13, line 306 - Mention the role of "infrastructures and waterways." Does the model consider stormwater infrastructure?
RESPONSE: We added a sentence into the text in Section 4.0 to clarify that we applied only drainage capacity assumptions and not specific details on stormwater infrastructure.

Figure 4 - Is the middle map of a different location? Why not use the same location for the pluvial flood mechanism as well?
RESPONSE: We have adjusted the middle map so that it represents the same area as the top and bottom map. We chose a different area originally to highlight an area of high pluvial

flood exposure but we agree that selecting the same area for all three maps may be more useful.

Pg. 19, line 32 - Word choice of the word "greater." Consider changing the first instance to "larger amounts of water" and the second instance to "further exposure" to add clarity.
RESPONSE: We have adopted the suggested wording.

Figure 5 - Consider splitting the classification of "continued exposure" into two groups: "continued exposure" and "worsened exposure" based on a building going from moderate to severe exposure. This could add more information to the figure to show not only where new exposure occurred but also where it was made worse by climate change.
RESPONSE: We agree with the suggestion and have taken it one step further. We divided the "continued exposure" into three groups (for comprehensiveness): (A) lessened exposure, (B) similar exposure, and (C) worsened exposure, as it is possible for a continued (but lower depth) exposure option as well. This has been reflected in Table 8 and the description before it. It was not, however, graphically displayed in the map for Figure 4 as it added complexity that took away from the main purpose of the graphic showing the higher-level categorizations of exposure.

Page 20, line 41 - What model assumptions are being referred to? Be specific.
RESPONSE: With the new revision, greater detail was added to the flood hazard modeling procedure in Section 3.2, "Research Methods". These hopefully clarify many of the input modeling assumptions that go into the hazard data, including under climate change conditions. Additionally, more detail was added in situ, particularly, the expectation that all major types of flooding, including fluvial (riverine), pluvial (rainfall) and storm surge (coastal) will intensify as the climate changes and the similar assumptions embedded into the JBA climate change modules involving precipitation, temperature, and run-off.

Page 20, line 43 - "two of the three assets." The sentence before mentions 14 buildings, so not sure where three assets are coming from.
RESPONSE: We changed the sentence to "Figure 6 illustrates 2 of the 14 assets that were classified as …"

Page 20, line 46 - the small pockets of disjointed water causing changes. Perhaps this is an opportune place to discuss the role of model uncertainty on the output since the only input that was changed was increasing rainfalls based on climate change.
RESPONSE: We added some discussion on the role of model uncertainty on the output here. These very small artifacts in map output (e.g. where we are seeing flooding in the future map that wasn't present in the non-climate change state) do reflect the complexity in the modelling process, and for example, the way rounding of parameters can propagate through to the end map slightly differently.

Page 21, lines 54-56 - can add that pluvial flooding was the driving mechanism, so excluding this in flood mapping could result in harmful consequences
RESPONSE: We added a sentence explaining the consequences that excluding pluvial flooding could have on flood mapping.

Page 21, line 61 - Provide more detail on the approaches' scalability.

RESPONSE: We have expanded on the scalability of the approach, notably the ability to simply substitute the flood hazard and exposure data of interest into the established code, and the analysis can be re-run in the new setting (p. 24).

Page 21, line 62 - Why is there greater uncertainty in the pluvial flood estimation?
RESPONSE: We have added detail on the greater uncertainty in pluvial flood estimation, notably that most regulatory mapping to compare against is fluvial (for calibration purposes), and nuanced engineered drainage systems are not widely available nor public (p. 24).

Page 22, line 83 - make 'building' plural
RESPONSE: We changed "building" to "buildings".

Page 22, line 84 - Add clarity to wording. Buildings are not a source of exposure, but an increase in development leads to a greater density and number of buildings that can be exposed to flooding.
RESPONSE: We revised the sentence to clarify that future development will lead to higher risks under future climate conditions because of the anticipated added density and number of buildings that will be exposed.

Page 22, line 85 - Again, buildings are not a source of flooding, but they increase the number of assets that can be exposed.
RESPONSE: We deleted the sentence because the previous sentence was revised to explain that future development will lead to higher risks because of the added density and number of buildings that will be exposed to flooding.

Page 22, lines 95-96 - Sentence is unclear.
RESPONSE: We clarified that while the model helps produce high resolution maps, further physical modelling could help to improve the precision to a 1- to 2-metre resolution.

Page 22, line 98: Validation is discussed, but what about the roles of uncertainty of and sensitivity analysis on the dataset and the results?
RESPONSE: We have added in a discussion on the role of uncertainty on the dataset and results in the Discussion (Section 5.0).

Page 24, line 161 - Reference is in the reference list but not cited in the manuscript
RESPONSE: This reference was removed.

Page 27, line 236 - Reference is in the reference list but not cited in the manuscript
RESPONSE: We adjusted the Pörtner reference in the reference list to IPCC (2019) – now included in the reference list.

**Reviewer 2**

**1. I agree with the first reviewer that the level of novelty in this paper is uncertain.**

RESPONSE: We have expanded on the novelty of the paper in the discussion and conclusion sections, noting that this is the first study that effectively maps fluvial, pluvial, and storm surge flood exposure based on climate change at a 5-metre in Vancouver, Canada, and while other models like FLO-2D, First Street Foundation, and Fathom do exist, no such model has covered Canada until now. We have conducted an analysis using JBA's Canada Flood Map,

which is the most widely used map in the Canadian insurance market, providing comprehensive coverage for fluvial, pluvial and coastal flooding.

**2. The authors claim that their primary focus is on a general flood exposure approach, but upon examination, it becomes evident that 90% of the content is around the flood hazard model. The introduction and literature review sections primarily delve into flood hazard modeling techniques and approaches. While a flood exposure method is presented in the methodology section, it appears to be quite limited.**

RESPONSE: We have added more information on flood exposure methods to the methodology section.

**3. The flood exposure methodology introduced in this paper is notably simple, and the level of uncertainty associated with it is high. In line 264, it is explained that when any part of a building is implicated by flood hazard data, the maximum flood depth value is assigned. Given the localized nature of the analysis, I would recommend that the authors explore alternative approaches to reduce this uncertainty.**

RESPONSE: We have added more discussion on this in the comment mentioned above, located in the methodology section.

**4. Furthermore, the authors make certain assumptions, such as those presented in line 279 where they introduce subclasses of exposure. While they argue that these assumptions are reasonable for Canada, they neither provide a comprehensive explanation nor support them with relevant literature.**

RESPONSE: We have added more information to this section – thank you.

**4. In light of these concerns, I would recommend major revisions to the paper. The primary objective of the paper should be clearly defined, and appropriate methodologies and approaches should be employed accordingly. There is a substantial body of literature available on flood exposure and local-level methods, which the authors should draw upon to enhance their work.**

RESPONSE: The primary objective of the paper has been defined towards the conclusion of the Introduction (Section 1.0), and the description of the methods was expanded in Section 2.0. Moreover, the literature review was expanded to discuss the state of physically-based models, which was then reflected on in the discussion section of this paper.